# Generation of a Novel High-Affinity Antibody Binding to PCSK9 Catalytic Domain with Slow Dissociation Rate by CDR-Grafting, Alanine Scanning and Saturated Site-Directed Mutagenesis for Favorably Treating Hypercholesterolemia

**DOI:** 10.3390/biomedicines9121783

**Published:** 2021-11-27

**Authors:** Zhengli Bai, Menglong Xu, Ying Mei, Tuo Hu, Panpan Zhang, Manman Chen, Wenxiu Lv, Chenchen Lu, Shuhua Tan

**Affiliations:** Jiangsu Key Laboratory of Druggability of Biopharmaceuticals, State Key Laboratory of Natural Medicines, Department of Molecular Biology, School of Life Science and Technology, China Pharmaceutical University, Nanjing 210009, China; baizhenglicpu@163.com (Z.B.); 1731030118@stu.cpu.edu.cn (M.X.); 18796622329@163.com (Y.M.); tuo_hu@163.com (T.H.); 1831030123@stu.cpu.edu.cn (P.Z.); chen13441@163.com (M.C.); 18779882494@163.com (W.L.); luchenchencpu@yeah.net (C.L.)

**Keywords:** single-chain variable fragment (scFv), alanine scanning, saturated site-directed mutagenesis, PCSK9, molecular docking, slow dissociation rate, catalytic domain

## Abstract

Inhibition of proprotein convertase subtilisin/kexin type 9 (PCSK9) has become an attractive therapeutic strategy for lowering low-density lipoprotein cholesterol (LDL-C). In this study, a novel high affinity humanized IgG1 mAb (named h5E12-L230G) targeting the catalytic domain of human PCSK9 (hPCSK9) was generated by using CDR-grafting, alanine-scanning mutagenesis, and saturated site-directed mutagenesis. The heavy-chain constant region of h5E12-L230G was modified to eliminate the cytotoxic effector functions and mitigate the heterogeneity. The biolayer interferometry (BLI) binding assay and molecular docking study revealed that h5E12-L230G binds to the catalytic domain of hPCSK9 with nanomolar affinity (*K*_D_ = 1.72 nM) and an extremely slow dissociation rate (*k*_off_, 4.84 × 10^−5^ s^−1^), which interprets its quite low binding energy (−54.97 kcal/mol) with hPCSK9. Additionally, h5E12-L230G elevated the levels of LDLR and enhanced the LDL-C uptake in HepG2 cells, as well as reducing the serum LDL-C and total cholesterol (TC) levels in hyperlipidemic mouse model with high potency comparable to the positive control alirocumab. Our data indicate that h5E12-L230G is a high-affinity anti-PCSK9 antibody candidate with an extremely slow dissociation rate for favorably treating hypercholesterolemia and relevant cardiovascular diseases.

## 1. Introduction

Hypercholesterolemia with elevated plasma low-density lipoprotein cholesterol (LDL-C) levels is a major risk factor for cardiovascular diseases (CVDs) [1,2]. Accumulating experimental and clinical studies implicate that a high level of LDL-C in the serum is closely associated with the incidence of CVDs [3,4,5]. Plasma LDL-C is mainly cleared through binding to the low-density lipoprotein receptor (LDLR) on the surface of hepatocytes, and the formed LDL-C/LDL-R complex is subsequently transported to lysosome for LDL-C degradation, and the LDLR is recycled to the cell surface [6].

However, it has been revealed that proprotein convertase subtilisin kexin type 9 (PCSK9) could mediate LDLR degradation by binding to the epidermal growth factor (A) and β-propeller domains of LDLR [7,8,9], thereby, PCSK9 has been now recognized as an attractive therapeutic target for hypercholesterolemia. Recently, several kinds of PCSK9 inhibitors, including small peptides [10], small interfering RNA (siRNAs) [11] and antibodies [12,13,14], have been developed. Of them, monoclonal antibodies (mAbs) that block the interaction of PCSK9/LDLR appear to be promising therapeutic agents to treat hypercholesterolemia because of their significant advantages such as being highly selective, well-tolerated and having fewer dosing frequencies [15].

The hybridoma technique, invented by Köhler and Milstein, is a well-established robust method for generating mAbs targeting the antigen of interest [16,17,18]. Yet, mAbs prepared by the hybridoma technique could induce the human anti-mouse antibodies (HAMA) response [19]. To circumvent this issue, multiple effective humanization strategies have been successfully developed [20]. Initially, chimeric antibodies were constructed by substituting the murine constant region with an appropriate human constant region to reduce the content of heterologous sequences [21], and several chimeric antibodies were approved by FDA, including ramucirumab, brentuximab, dinutuximab, etc. In chimeric antibodies, the variable domains that consist of framework regions (FRs) and complementarity determining regions (CDRs) remain to be completely murine [22]. Considering FRs are not necessarily required for antigen recognition, grafting only the murine CDRs to the human FRs has been demonstrated as an excellent method to further reduce the immunogenicity of chimeric antibodies [20,23]. Up to now, quite a number of CDR-grafted antibodies, such as daclizumab, trastuzumab and palivizumab, etc., have been granted marketing approval [24].

However, grafting murine CDRs onto the FRs of human antibodies often leads to a lower affinity for the target antigen and requires additional processes to restore or improve the affinity [24]. Back-mutation of some residues in human FRs to the original murine ones was commonly applied to rescue the antibody’s binding affinity. In addition, in vitro affinity maturation of mAbs, which mimics the natural affinity maturation in vivo [25], could also help to restore the affinity of humanized antibodies. The most commonly used in vitro affinity maturation technologies are (1) site-directed mutagenesis and random mutagenesis consisting in the introduction of mutagenesis specifically or randomly throughout the gene, (2) chain shuffling that recombines the heavy and light chains of different antibodies with high affinity, (3) phage display which allows screening the desired antibody from a large library with millions to trillions of variants [26,27,28].

In the present study, we aimed to generate a high-affinity hybridoma-derived mAb against hPCSK9 for future therapeutic applications. In order to reach this goal, a CDR-grafting humanization approach was initially performed to reduce the immunogenicity of selected murine mAb, some key residues in murine FRs that might influence the antigen-binding activity were back-mutated attempting to restore full affinity. Thereafter, alanine-scanning mutational analysis followed by a saturated site-directed mutagenesis process were conducted on the critical amino acid residue of humanized antibody to further improve its affinity. Finally, the selected optimized scFvs were reformatted into full-length IgG by fusing with the modified human IgG1 constant region for favorably treating hypercholesterolemia in vivo.

## 2. Materials and Methods

### 2.1. Materials

MEM, DMEM, Opti-MEM, and Pluronic-F68 were obtained from Thermo Fisher Scientific (Waltham, MA, USA). Fetal bovine serum (FBS), penicillin G sodium salt, streptomycin solution, and hypoxanthine-aminopterin-thymidine (HAT) were obtained from MilliporeSigma (Burlington, MA, USA). HyClone™ HyCell™ CHO Medium was bought from GE Healthcare (Piscataway, NJ, USA). Additionally, 25 kDa linear polyethyleneimine (LPEI) was obtained from Polysciences (Warrington, Pennsylvania, USA). Quickantibody-Mouse 5W adjuvant (Cat# KX0210041) was obtained from Biodragon Immunotechnologies (Beijing, China). Bovine serum albumin (BSA) was purchased from Biofroxx (Einhausen, Hessen, Germany). Agarose Gel DNA Extraction Kit, RNAiso Plus, and PrimeScript RT Reagent Kit with gDNA Eraser were bought from TaKaRa (Dalian, Liaoning, China). Rabbit anti-LDLR antibody (Cat# ab52818) and rabbit anti-PCSK9 antibody (Cat# ab181142) were bought from Abcam (Cambridge, UK). Glutamine, TMB substrate, IPTG, HRP-conjugated goat anti-rabbit IgG (Cat# D110058), rabbit anti-GAPDH antibody (Cat# D110016), and Alexa Fluor 488^®^-conjugated goat anti-rabbit IgG (Cat# D110061) were bought from Sangon Biotech (Shanghai, China). DiI-LDL was obtained from Yiyuan Biotechnologies (Guangzhou, Guangdong, China). Commercial test kits for LDL-C, TG, TC, and HDL-C were obtained from Jiancheng Bioengineering Institute (Nanjing, Jiangsu, China).

### 2.2. Bacterial Strains and Cell Lines

*Escherichia coli* (*E. coli*) strains DH5α and BL21 (DE3) were used as hosts for plasmid preparation and single-chain variable fragment (scFv) prokaryotic expression, respectively. Chinese hamster ovary (CHO-3E7) cells were obtained from Genscript Biotech (Nanjing, China), cultured in HyClone™ HyCell™ CHO Medium, and used as hosts for IgG1 eukaryotic transient expression. Mouse myeloma cell line SP2/0 was purchased from American Type Culture Collection (ATCC, Manassas, VA, USA) and cultured in DMEM medium supplemented with penicillin (100 U/mL), streptomycin (100 μg/mL) and 10% (*v*/*v*) FBS. Human hepatic HepG2 cells were obtained from China Infrastructure of Cell Line Resources (Beijing, China) and maintained at 37 °C, 5% CO_2_, in MEM supplemented with 10% (*v*/*v*) FBS, penicillin (100 U/mL) and streptomycin (100 μg/mL). All cells were cultured in a humidified incubator at 37 °C in an atmosphere of 5% CO_2_ in the air.

### 2.3. Antigen Preparation

The recombinant plasmids expressing hPCSK9 (pTT5-hPCSK9) were constructed as previously described [29], and transiently transfected into suspension CHO-3E7 mammalian cells using PEI transfection reagent [30]. On day 7 post transfection, the supernatant was purified with a Ni^2+^-based immobilized metal ion affinity chromatography (Ni-IMAC, GE Healthcare, Piscataway, NJ, USA), followed by Superdex^TM^ 200 HR 10/300GL size-exclusion chromatography (GE Healthcare, Piscataway, NJ, USA). Protein concentration was determined using the BCA protein assay kit (Beyotime, Shanghai, China).

### 2.4. Generation of Murine Monoclonal Antibodies

Female BALB/c mice (6–8-week, Qinglongshan Experimental Animal Breeding Farm, Nanjing, Jiangsu, China) were intramuscularly immunized with adjuvant-combined hPCSK9 protein (20 μg per mouse) on day 1 and day 21. The final boost (50 μg of hPCSK9 protein) was given intraperitoneally on day 35 without adjuvant. Three days after the final booster immunization, orbital blood of mice was collected for antibody titer detection. When the antibody titters attained 1:100,000, the cell fusion was conducted according to standard cell fusion procedures [16,31]. Afterward, the positive hybridoma cells were screened by indirect ELISA and subcloned three times by limiting dilution method. The ascites of identified hybridoma were also prepared by injection of 1 × 10^6^ positive hybridoma cells into the peritoneal cavity of pristine-treated BALB/c mice, and the ascites containing specific mAbs were purified by protein A affinity chromatography (Roche, Mannheim, Germany). The isotype of purified mAb was determined using a mouse monoclonal subtype identification kit (KMI-2, ProteinTech Group, Chicago, IL, USA) according to the manufacturer’s instruction.

### 2.5. Enzyme-Linked Immunosorbent Assay

The hybridoma cells producing antibodies against hPCSK9 were screened by enzyme-linked immunosorbent assay (ELISA) using 96-well plates coated with hPCSK9 (1 μg/mL) in coating buffer (0.2 M Na_2_CO_3_/NaHCO_3_, pH 9.6) overnight at 4 °C. The plates were then blocked with PBS containing 3% (*w*/*v*) BSA for 2 h at 37 °C and incubated with 100 μL of hybridoma supernatants at 37 °C for 2 h. Additionally, non-competitive phage ELISA with the addition of increasing concentrations of mAb (10^−^^1^, 10, 10^2^, 10^3^, 10^4^, 10^5^ ng/mL) was also set up to further measure the affinity constant (K_aff_) of selected mAb as described previously [32]. After washing three times with 0.1% Tween in PBS (PBST), HRP-conjugated goat anti-mouse IgG antibody was added and incubated for 1 h at 37 °C. Finally, the TMB substrate was added and allowed to develop for 15 min at room temperature, and the absorbance at 450 nm was detected using a microplate reader (Thermo Scientific, Waltham, MA, USA).

### 2.6. Western Blot Analysis

The cells or tissues lysates were homogenized in RIPA lysis buffer and collected by centrifugation at 12,000× *g* as previously described [33]. After protein concentrations determination with BCA protein assay, equal amounts of samples were subjected to 12% (*w*/*v*) SDS-PAGE and transferred to 0.22 μm polyvinylidene fluoride (PVDF) membrane (MerckMillipore, Darmstadt, Germany). After blocking with 0.1% (*v*/*v*) TBS-Tween 20 (TBST) containing 5% (*w*/*v*) nonfat milk for 2 h at room temperature, the membrane was incubated with primary antibodies against LDLR (Cat# ab52818, 1:1000) or GAPDH (Cat# D110016, 1:1000) at 4 °C overnight, and then incubated with HRP-conjugated goat anti-rabbit IgG (Cat# D110058, 1:5000) at 25 °C for 1.5 h. Protein bands were visualized by enhanced chemiluminescence (ECL) solution (Thermo Scientific, Waltham, MA, USA) and quantified with the ImageJ 1.42q (National Institutes of Health, Bethesda, MD, USA).

### 2.7. LDL-C Uptake Assay

LDL-C uptake assay was conducted as previously described [34] with slight modification. In brief, HepG2 cells were seeded in black 96-well plates at a density of 3 × 10^4^ cells per well and cultured overnight. Cells were then pretreated with opti-MEM for 12 h, and treated with 20 μg/mL hPCSK9 alone or co-treatment with 50 μg/mL antibodies for 8 h. Afterwards, 20 μg/mL DiI-LDL was added to each well and incubated for an additional 4 h. After washing 3 times with PBS in the dark, LDL-C uptake levels were measured using a Multimode Reader (Thermo scientific, Waltham, MA, USA) at 520 nm excitation/580 nm emission.

### 2.8. Cloning of VH and VL Gene from Hybridoma Cells

Total RNA was isolated from hybridoma cells secreting monoclonal antibody (mAb) against hPCSK9 by RNAiso reagent and quantified by measuring A_260_ nm on a Thermo NanoDrop 2000 (Thermo Fisher Scientific, Waltham, MA, USA). Then the first-strand cDNA was amplified by reverse transcription-polymerase chain reaction (RT-PCR) using PrimeScript™ RT reagent Kit with gDNA Eraser (TaKaRa, Dalian, China), and the variable region genes of the heavy and light chains of selected mAb were respectively amplified by PCR using PrimeSTAR^®^ HS DNA Polymerase (TaKaRa, Dalian, Liaoning, China) and previously published primer pairs [35] with minor modification (Appendix A).

### 2.9. Computer Modeling of Single-Chain Variable-Fragment Antibodies

The three-dimensional (3D) structure models of single-chain variable-fragment antibodies (scFvs) were built via homology modeling using the Schrodinger Suite 2009 (Schrödinger, LLC, New York, NY, USA). The Gromacs program, a versatile package in Schrodinger software to perform molecular dynamics, was used to optimize the structure of m5E12scFv in silico and make it closer to the conformation in the natural environment Subsequently, the qualities of the constructed models were evaluated by Ramachandran Plot within Discovery Studio software v2.5 (Accelrys Software Inc., San Diego, CA, USA). Root mean square deviation (RMSD) is a statistic to assess the deviation degree between resulting and target conformations, which here was used to estimate the model deviation of the constructed models. Thus, the RMSD values of the main chain atoms (C-α, C, N, O) in CDRs between the models were also calculated by Molecular Operating Environment (MOE) software (Chemical Computing Group, Montreal, Quebec, Canada).

### 2.10. Design of Humanized scFvs

Two humanized scFvs were designed based on different antibody humanization methods. The first humanized scFv was designed by transferring murine CDRs onto suitable human consensus FR templates with the highest similarity using a traditional approach called CDR grafting [36]. In order to maintain murine CDRs conformation, the key residues, including (a) residues with less than 30% surface accessibility [37] (b) abnormal residues [38], (c) “Vernier” residues located at the FR-CDR junction [39], which may change CDRs conformations were screened by Abcheck (https://www.abcheckantibodies.com/ accessed on 27 October 2017) and Schrodinger Suite 2009. Afterward, another humanized scFv was then designed by back-mutation of these key residues to the amino acids of the original murine mAb.

### 2.11. Construction, Expression, and Purification of scFvs

The genes encoding the humanized scFvs were synthesized at Genscript Biotech and subcloned into pET-27b (Novagen, Madison, WI, USA) containing the pectate lyase signal peptide (pelB) of Erwinia carotovora using primers listed in Appendix A. The resulting recombinant plasmids were further transformed into *E. coli* BL21 (DE3) cells through CaCl_2_ heat shock method, and the transformed *E. coli* BL21 (DE3) cells were cultured in 2 × YT medium with 50 μg/mL of kanamycin at 37 °C. When the OD_600_ reached 0.6–0.8, the temperature was shifted to 16 °C and 0.2 mM isopropyl-β-galactosidase (IPTG) was added to induce expression for 18 h. The expressed scFv proteins with C-terminal 6 × His-tagged were isolated from the periplasm and purified by Ni-NTA affinity chromatography column (GE Healthcare, Piscataway, NJ, USA) and Superdex^TM^ 75 HR 10/300GL size exclusion (GE Healthcare, Piscataway, NJ, USA) successively, according to manufactures’ protocols.

### 2.12. Saturated Site-Directed Mutagenesis of Humanized scFv

The residues in HCDR3 and LCDR3 are most likely to dominate the antibody–antigen interaction [40]. To further identify critical residues involved in hPCSK9 binding, alanine scanning mutagenesis was carried out in these two regions by one-step PCR technology [41]. Briefly, residues in HCDR3 and LCDR3 excepting (a) Tyr and Trp which are advantageous for large van der Waals or hydrophobic interactions, (b) Asn and Ser which mainly form hydrogen bonds, (c) glutamine at the 89th and 90th position of the light chain [42], were selected to mutate to alanine. The oligonucleotide primers used were listed in Appendix A. The effect of each mutated site on the activity of humanized scFv was determined by measuring the changes in LDL uptake after treating HepG2 cells with purified humanized scFv proteins. Afterward, the key residues were identified and site-directed saturation mutagenesis was conducted on these residues using the primers listed in Appendix A.

### 2.13. Generation of Full-Length Antibodies

To generate full-length antibodies, the V_H_ and V_L_ of humanized scFvs were fused with the constant region of heavy chain (HC) and kappa light chain of human IgG1 (LC, Accession number: ABU90709.2) by overlap-extension PCR (OE-PCR) [43], respectively. Primers used for OE-PCR were listed in Appendix A. In order to eliminate antibody-dependent cellular cytotoxicity (ADCC) and complement-dependent cellular cytotoxicity (CDC) effects, the human IgG1 constant region was modified with L234A/L235A/N297G [44,45,46], and the C-terminal lysine residue in HC was deleted to reduce heterogeneity [47]. The amplified HC and LC coding genes (GenBank accession number: MW715631, MW715632, MW725291, MW725292, MW725293) containing a signal peptide sequence (‘MDWTWRFLFVVAAATGVQS’ for HC secretory expression, Accession number: CAA34971.1; ‘MDMRVPAQLLGLLLLWLSGARC’ for LC secretory expression, Accession number: S24320) at the 5′-end were then subcloned into a eukaryotic expression vector pTT5, respectively. The constructed recombinant plasmids (Appendix A) were co-transfected into CHO-3E7 mammalian cells at a 1:1 ratio (*w*:*w*) [30], and the culture supernatants were purified with a protein A column (Roche, Mannheim, Germany).

### 2.14. Binding Affinity Measurement

ForteBio Octet QK^e^ (ForteBio, Fremont, CA, USA), a biomacromolecule interaction analysis system, was used to measure the affinity constants of antibodies. In short, hPCSK9 (50 μg/mL) was biotinylated at room temperature for 2 h using a biotinylation kit (Genemore, Shanghai, China) and immobilized on the surface of streptavidin biosensors (ForteBio) for 300 s. The antigen-captured biosensors were then dipped into series 2-fold dilution of antibodies for 300 s or longer and moved to SD buffer (PBS containing 0.02% Tween 20 and 0.1% BSA, pH 7.4) without antibodies for 600 s. The concentrations of scFvs were 6000, 3000, 1500, 750, 375, 187.5, 93.75 nM, and mAbs were 800, 400, 200, 100, 50, 25, and 12.5 nM. The kinetic constants including k_on_, k_off_, and K_D_ were analyzed by using ForteBio data analysis software v7.1 (ForteBio, Fremont, CA, USA).

### 2.15. Homology Modeling and Protein Contact Identification

To explore the specific binding mechanism of the antigen-antibody, the three-dimensional (3D) structure model of the Fab fragment of h5E12-L230G was constructed by SWISS-MODEL Workspace (http://swissmodel.expasy.org/ accessed on 10 March 2021) based on the top-ranked template with known structure, and the stereochemical property was checked through the Ramachandran plot [48]. Subsequently, the generated models were further docked with a high-resolution 2.3 Å crystal structure of PCSK9 (PDB ID:5OCA) using the BioLuminate module of Schrödinger Software Suite 2009 (Schrödinger). Following this, the key interaction residues between the antibody and PCSK9 were identified using the Pymol software Version 2.3.0 (Schrödinger), and the free binding energy (△G_binds_) of the Fab-PCSK9-complex were calculated using the Molecular Mechanics/GB Surface Area (MM/GBSA) method (http://cadd.zju.edu.cn/hawkdock/ accessed on 12 March 2021) [49].

### 2.16. Studies in Mice

Male C57BL/6 mice aged 6–8 weeks were obtained from Qinglongshan Experimental Animal Breeding Farm (Certificate no. SCXK (Su) 2017-0001; Nanjing, Jiangsu, China) and housed in a temperature (25 °C) and humidity-controlled (50–60%) room with access to food and water ad libitum. Following 1 week of acclimation, mice were randomly split into 8 groups (a normal group, a model group, and six treatment groups, *n* = 6 per group). On day 1, the model group and dosing group were injected 2 mL saline containing 50 μg pTT5-hPCSK9 intravenously in 5–7 s to establish hyperlipidemic mouse model [50,51], while the normal group was just injected with 2 mL saline. On day 7, the dosing groups were administered with 1, 3, and 10 mg/kg of mAbs in 100 μL saline, respectively, while the normal and model groups were administered same-volume saline. Then the mice were fasted for 8 h and euthanized for blood sample collection. Liver tissues were also collected, and dissected into two parts, one was homogenized by RIPA buffer containing 1 mM PMSF for Western blot analysis, the other part was fixed with 4% (*w*/*v*) paraformaldehyde for 48 h and routinely embedded in paraffin for immunofluorescence analysis. All experiments were performed under a project license (NO.: 201601179) granted by the Animal Ethics Committees of China Pharmaceutical University, in compliance with national guidelines for the care and use of animals.

### 2.17. Immunofluorescence Analysis

Immunofluorescence staining was performed to detect LDLR protein levels in mice livers as previously described [33] with minor modification. Briefly, after deparaffinization and hydration, liver tissue sections were pretreated by heating for 20 min in boiling sodium citrate solution (0.01 M, pH 6.0) for antigen retrieval. Afterward, the tissue sections were blocked with 10% (*v*/*v*) goat serum for 1 h at room temperature and incubated with rabbit anti-LDLR antibody (Abcam,1:100) at 4 °C overnight. After washing three times with PBS, the sections were incubated with Alexa Fluor^®^ 488-conjugated goat anti-rabbit IgG (Abcam, 1:200) at 37 °C for 1 h in the dark and the stained sections were mounted with a drop of glycerin. Images were taken under a Zeiss AX10 fluorescence microscope (Zeiss, Oberkochen, Germany).

### 2.18. Statistical Analysis

Data are expressed as the mean ± SEM of multiple experiments. Comparison between groups was performed using one-way analysis of variance followed by a Tukey multiple comparison test with GraphPad Prism v5.0 software (San Diego, CA, USA). Results were considered significant when *p*-values < 0.05.

## 3. Results

### 3.1. Generation of Murine Mab against Hpcsk9 by Hybridoma Technology

To generate murine mAb against hPCSK9, the recombinant hPCSK9 protein as the antigen was expressed by transient transfection of the pTT5-hPCSK9 plasmid (Figure 1A) to CHO 3E7 cells and purified by Ni-IMAC and size-exclusion chromatography as described above. Purified hPCSK9 was then identified by 10% (*w*/*v*) SDS-PAGE (Figure 1B) under reducing condition and Western blot (Appendix A) using the rabbit anti-human PCSK9 antibody (Abcam, 1:3000).

Subsequently, BALB/c mice were immunized with purified hPCSK9 protein. On day 38, the mice attaining antibody titer of 1:640,000 (Figure 1C) were sacrificed, and the splenocytes were fused with SP2/0 cells for hybridoma production. A positive hybridoma clone, named 5E12, was identified by ELISA (Figure 1D) and subcloned three times by limiting dilution. This hPCSK9-specific murine antibody (named m5E12) was then purified from mouse ascites by protein A affinity chromatography (Roche, Mannheim, Germany) and identified by SDS-PAGE under reducing and nonreducing conditions (Appendix A). The purified m5E12 was analyzed by Shodex PROTEIN KW-802.5 (SHOWA DENKO K.K., Japan) showing a purity of 99% (Appendix A).

### 3.2. Characterization of Generated m5E12

Firstly, we determined the isotype of m5E12 using a commercial murine antibody isotyping kit (KMI-2, ProteinTech Group, Chicago, IL, USA) according to the manufacturer’s instructions. The result showed that m5E12 belonged to the subtype IgG1 and the light chain of the mAb was kappa (Appendix A). Secondly, the specificity and affinity (K_aff_) of m5E12 to hPCSK9 were analyzed by Western blot and ELISA. The results revealed that hPCSK9 protein could be specifically recognized by m5E12 (Figure 2A) and the affinity constant (K_aff_) of m5E12 to hPCSK9 protein was 1.04 × 10^9^ M^−^^1^ (Figure 2B). Finally, we tested the effects of m5E12 on the levels of LDLR and LDL-C uptake in HepG2 cells. As shown in Figure 2C,D, m5E12 effectively increased LDLR levels and promoted the LDL-C uptake in HepG2 cells as compared to the hPCSK9-treated group.

### 3.3. Humanization of Murine 5E12 scFv (m5E12scFv)

For antibody humanization, the VH (Figure 3A) and VL (Figure 3B) amino acid sequences of m5E12 were determined by RT-PCR and gene sequencing. The VH and VL of m5E12 were then linked in a format of VH-(Gly4Ser)3-VL, named m5E12scFv. Afterwards, humanization of m5E12scFv was accomplished by CDR grafting without (named h5E12scFv) or with back mutation (h5E12scFv-bm) by modeling.

In detail, one of the humanized variable fragments, named h5E12scFv, was designed by grafting the CDRs of m5E12 onto the heavy chain (GenBank accession number: AMK70123.1) and light chain (GenBank accession number: APZ85158.1) of human antibodies (Figure 3A,B). Back-mutation was then performed on h5E12scFv. On the one hand, N97 at the heavy chain and E45, R63, V78 at the light chain of the two human templates were rare residues and were mutated to corresponding conserved residues. On the other hand, the three-dimensional (3D) structures of m5E12scFv (Figure 3C), h5E12scFv (Figure 3D), and h5E12scFv-bm (Figure 3E) were modeled by Schrodinger software based on crystal structure with the highest sequence identity and verified by Ramachandran plot (Appendix A), and nine key residues (E1, K38, I48, K67, A68, V72, A79, L81, and S92) and seven key residues (I4, T8, I46, H59, E85, F87, and E100) in FRs of murine VH and VL, which may change CDRs conformations, were also back-mutated (named h5E12scFv-bm).

The RMSD value for the three modeled entire structures (Figure 3C–E) was 0.724 Å. The RMSDs for five non-HCDR3 loops ranged from 0.255 Å to 0.523 Å, and the HCDR3 RMSD was 1.150 Å. Since all RMSDs were less than 1.5 Å, the three structures were considered to share the same conformation at the computer level.

### 3.4. Preparation and Selection of Humanized 5E12 scFv

To prepare humanized 5E12 scFv proteins for selection, the genes encoding m5E12scFv, h5E12scFv, and h5E12scFv-bm fragments were synthesized at Genscript Biotech (Nanjing, China), amplified by PCR (Appendix A) and then inserted into T7 promoter driven expression vector pET-27b between *Nco* I and *Hin*d III sites (Appendix A). The construct was transformed in *E. coli* BL21 (DE3) cells as described. After induction with 0.2 mM IPTG at 16 °C for 18 h, humanized 5E12 scFv proteins (Appendix A) were purified by Ni-NTA affinity chromatography column.

The specificity and kinetic parameters of purified humanized 5E12 scFvs binding to hPCSK9 were further determined by Competitive ELISA and ForteBio Octet QK^e^ System. As shown in Appendix A, both h5E12scFv and h5E12scFv-bm could competitively react with hPCSK9, but the binding ability of h5E12scFv-bm was relatively weaker than h5E12scFv. It was further observed by BLI (Appendix A) that h5E12scFv exhibited the highest affinity (*K*_D_ = 1.71 × 10^−7^ M) to hPCSK9 with slower dissociation rate (*k*_off_ = 1.08 × 10^−3^ s^−1^) than that of m5E12scFv (*k*_off_ = 7.44 × 10^−3^ s^−1^) and h5E12scFv-bm (*k*_off_ = 1.00 × 10^−2^ s^−1^), which is related with the lifetime of the antibody-antigen complex [52,53].

Additionally, we tested the effects of humanized 5E12 scFvs on the LDLR and LDL-C uptake levels in HepG2 cells. As shown in Figure 4A,B, h5E12scFv potently elevated the levels of LDLR and enhanced the LDL-C uptake in HepG2 cells as compared to the hPCSK9 group, but there is still a certain gap (Figure 4B) compared with the scFv form of Alirocumab (named Ali-scFv).

### 3.5. Affinity Maturation of h5E12scFv In Vitro

To further enhance the affinity and bioactivity of h5E12scFv, we firstly identified the critical residues by alanine-scanning mutagenesis of several residues including F99, H100, D102, D104, F106, D107, R227, P229, L230, T231, respectively, and 10 mutants (Figure 5A) named h5E12scFv-F99A, h5E12scFv-H100A, h5E12scFv-D102A, h5E12scFv-D104A, h5E12scFv-F106A, h5E12scFv-D107A, h5E12scFv-R227A, h5E12scFv-P229A, h5E12scFv-L230A, and h5E12scFv-T231A were purified (data not shown). The biological activity of these mutants was compared by measuring the changes of LDL uptake using HepG2 cell-based assay. The results (Figure 5B) showed that h5E12scFv-D102A and h5E12scFv-D107A exhibited scarcely any PCSK9 inhibitory effect, indicating that D102 and D107 was an essential residue for maintaining h5E12scFv’ activity and should be retained. Furthermore, h5E12scFv-L230A displayed stronger biological activity than parental antibody h5E12scFv, suggesting that the mutation of L230 to other residues might improve the hPCSK9 inhibitory effect of h5E12scFv.

Secondly, site-saturated mutagenesis experiments were carried out on the L230 residue of h5E12scFv, and the corresponding mutants (Appendix A) were purified and screened by LDL uptake assay (Figure 5C). It was shown that the LDL uptake levels were effectively enhanced by h5E12scFv-L230A, h5E12scFv-L230S, and h5E12scFv-L230G, and the LDL uptake levels in these three groups were restored to the comparable levels as in the Ali-scFv group. Thus, we chose h5E12scFv-L230A, h5E12scFv-L230S, and h5E12scFv-L230G for further construction of the full-length antibodies and in vivo functional studies.

### 3.6. Generation and Characterization of Full-Length Anti-PCSK9 Antibodies

The full-length format of anti-PCSK9 antibodies was constructed by connecting the V_H_ and V_L_ with a modified human IgG1 heavy-chain constant region and kappa light chain constant region, respectively. The heavy (GenBank accession number: MW715631) and light chain DNA sequence (GenBank accession number: MW715632, MW725291, MW725292, MW725293) of full-length antibodies were then inserted into the pTT5 vector (Appendix A) and co-transfected into CHO-3E7 cells for transient expression. After purification by protein A affinity chromatography columns, the obtained mAbs were verified by SDS-PAGE under non-reducing and reducing conditions (Figure 6A).

Subsequently, Dil-LDL uptake assay was performed as described above to test the hPCSK9 inhibitory effect of purified mAbs. The results (Figure 6B) showed that all the generated mAbs, including h5E12, h5E12-L230A, h5E12-L230S, h5E12-L230G, could significantly inhibit PCSK9-induced decrease in Dil-LDL uptake of HepG2 cells (*p* < 0.001). Among them, h5E12-L230G exhibited the most potent activity in enhancing the LDL-C uptake levels, which was comparable to the positive control alirocumab.

We further detect and compare the affinity constant of h5E12-L230G and alirocumab to hPCSK9 using the BLI method. As shown in Figure 6C,D and Appendix A, h5E12-L230G displayed a moderately slower association rate (k_on_= 2.81 × 10^4^ M^−1^s^−1^) and a slightly slower dissociation rate (k_off_ = 4.84 × 10^−5^ s^−1^) than alirocumab (k_on_= 8.04 × 10^4^ M^−1^s^−1^, k_off_ = 6.87 × 10^−5^ s^−1^), thus yielding a ~2-fold lower affinity (K_D_ = 1.72 × 10^−9^ M vs. 8.54 × 10^−10^ M) as compared to alirocumab. It can be concluded that the slower dissociation rate (k_off_) between h5E12-L230G and hPCSK9 results in a longer binding period, which may enhance the hPCSK9 inhibitory of h5E12-L230G to the comparable level of Alirocumab.

In addition, to further elucidate the interaction details at the paratope–epitope interface, the 3D structure of Fab region of h5E12-L230G (Appendix A) was built based on the top-ranked template crystal structure (PDB ID:6DW2). Ramachandran plot (Appendix A) of the modeled h5E12-L230G revealed that 97.91% of the residues were in the most favorable and allowed regions, indicating that the modeled structure was appropriate for the molecular docking analysis. The following docking results (Figure 6E) suggested that h5E12-L230G binds to the catalytic domain of hPCSK9, and both the heavy-chain and light-chain variable domain contributed to protein–ligand interaction. It appeared that as many as fourteen residues (S25, T28, W33, N55, H100, D102, Y103, D107, Y108 in heavy chain and Y49, S50, Y53, R54, S56 in light chain) in h5E12-L230G and up to fourteen residues (A168, L179, E181, E197, G198, R199, V200, V202, R237, D238, K243, S246, P279, S401 in catalytic domain) in hPCSK9 involved in the interactions, forming fifteen hydrogen bonds and two ionic bonds in the binding pocket (Appendix A). Finally, the binding free energy (ΔG_bind_) of the illustrated docking mode calculated using the MM-GBSA method was as low as −54.97 kcal/mol, indicating the h5E12-L230G showed high binding strength with hPCSK9.

### 3.7. Hypolipidemic Effect of h5E12-L230G in Mice Over-Expressing hPCSK9

We next evaluated the lipid-lowering efficacy of h5E12-L230G in vivo. The mice over-expressing hPCSK9 was established through HDD of 50 μg naked plasmid pTT5-hPCSK9. On day 6 after HDD, the mice in the treatment groups were given once by tail vein injection of h5E12-L230G. The results of Western blot (Figure 7A) and immunofluorescence (Figure 7C) revealed that h5E12-L230G dose-dependently up-regulated the LDLR levels in liver tissues 18 h after administration. Additionally, it was shown that treatment with h5E12-L230G at 1, 3, and 10 mg/kg resulted in a 5.8% (*p* > 0.05), 30.1% (*p* < 0.01), and 36.2% (*p* < 0.001) reduction in serum LDL-C relative to the model group, respectively (Figure 7B and Table 1), and h5E12-L230G treatment could also significantly lowered the levels of serum total cholesterol (TC) and triglyceride (TG), but did not significantly affect serum high-density lipoprotein (HDL-C) levels (Table 1).

## 4. Discussion

Elevated plasma LDL-C in patients with hypercholesterolemia is a well-established risk factor for CVDs. For over three decades, numerous clinical trials have firmly demonstrated the validity of LDL-C reduction in the prevention of CVD developing [4,54] and LDL-C levels have been accepted as a reliable efficacy endpoint for curative assessment and drug approval [55]. Nowadays, reduction of plasma LDL-C levels with the 3-hydroxy-3-methylglutaryl-coenzyme (HMG-CoA) reductase inhibitors (statins) remains the cornerstone of lipid management for the reduction of cardiovascular (CV) events in both primary and secondary prevention [56]. However, a significant percentage of patients cannot tolerate statin therapy due to the side-effect or cannot reach their recommended LDL-C goals even with a high enough statin dose [57,58]. Thus, it is necessary to develop novel LDL-C lowering drugs for these special patient populations.

PCSK9, a secreted protein able to induce the degradation of LDLR thereby reduce the clearance of LDL particles, has emerged as a promising target to reduce circulating LDL-C levels [15,59]. Nowadays, mAbs remain one of the most promising molecular targeted drugs because of their high specificity, and long half-life, and administration of PCSK9-targeting mAbs is an effective therapeutic way to suppress the interaction between PCSK9 and LDLR [60,61]. More recently, several mAbs targeting PCSK9, such as evolocumab, alirocumab [62], and LY3015014 [63], have been approved for clinical use or under clinic trials. In the present study, we chose alirocumab as the positive control and expected to generate a potent anti-hPCSK9 mAb with good druggability utilizing hybridoma-based methods.

Using the hybridoma technique, we initially generated a murine mAb targeting hPCSK9 (named m5E12), which could restore PCSK9-induced LDLR degradation and inhibit LDL uptake in HepG2 cells (Figure 2C,D). What needs illustration is that the humanization and affinity maturation of m5E12 here were performed using the scFv format instead of using the full-length antibody format—this is because scFv proteins can be easily obtained from *E. coli* expression system, which has the advantage of being fast-growing, inexpensive and easy to manipulate [64].

Specifically, to reduce the immunogenicity of mouse antibody, m5E12 was humanized by means of CDR grafting and back-mutation methods [36,37,38,39], generating two humanized m5E12 (h5E12scFv and h5E12scFv-bm) in the respective scFv format. Intriguingly, although the RMSD analysis showed that the humanized scFvs converged to the same conformation (Figure 3C–E), their affinity (*K*_D_) and biological activity were different. It was revealed that h5E12scFv displayed a higher affinity (*K*_D_) and hPCSK9 inhibitory effect than h5E12scFv-bm (Appendix A and Figure 4A,B). It has been reported that framework back-mutations help to maintain the native murine paratope conformation (57). However, in this report, the replacement of 16 key residues (Figure 3A,B) into those in the parent murine antibody failed to improve the affinity of CDR-grafted antibody (Appendix A). Moreover, it should be pointed out that the back-mutation processes also increase the non-human content which could increase the potential risk of immunogenicity in humans, revealing that the framework back-mutation might not always be needed in the development of murine-derived therapeutic antibodies.

To further improve the antibody’s affinity and bioactivity, alanine scanning mutagenesis was firstly conducted on the HCDR3 and LCDR3 of h5E12scFv to identify critical residues involved in antigen binding. Ten residues in HCDR3 and LCDR3 were selected to mutate to alanine, and we found that the L230A variant, termed h5E12scFv-L230A, was able to restore LDL uptake to the level slightly higher than that of parental antibody h5E12scFv (Figure 5B). Therefore, we assumed that the alteration of L230 residues in h5E12scFv might help to improve its activity, and fortunately, the following site-saturated mutagenesis experiment proved this hypothesis (Figure 5C). These results suggested that alanine-scanning mutational analysis along with a saturated site-directed mutagenesis process is a high-efficiency and powerful combined strategy to improve antibody activity, and mutagenesis of only one residue in LCDR3, in some cases, could obviously affect the property of an antibody.

To date, the majority of therapeutic antibodies developed are full-length IgG, and the full-length IgG molecules are considered one of the most suitable formats for clinical applications because of their increased stability, acquired multivalent binding, and extended half-life [65]. In addition, it is worth noting that therapeutic mAbs with cytotoxic effector function are not desirable for non-oncologic therapeutic applications. Therefore, the selected optimized scFv mutants (h5E12scFv-L230A, h5E12scFv-L230G, h5E12scFv-L230S) here were reformatted into full-length Fc-silenced IgG1 format which would not trigger ADCC and CDC effects [45,46]. Of these full-length antibodies, h5E12-L230G could reverse LDL uptake to the comparable levels of alirocumab (Figure 6B) and bind to hPCSK9 with a 1.41-fold slower dissociation rate (k_off_ = 4.84 × 10^−5^ s^−1^) than alirocumab (k_off_ = 6.87 × 10^−5^ s^−1^). It is notable that a targeted therapeutic drug with a slow dissociation rate appears to be effective since slow dissociation rate contributes to the long-lasting target binding of drugs [52]. Thus, the extremely slow dissociation rate of h5E12-L230G to hPCSK9 might partly account for its potent hPCSK9 inhibitory effect in vitro and in vivo.

It has been shown that both the catalytic domain and C-terminal domain targeting antibodies have inhibitory effects on the function of PCSK9 [66,67]. We have also previously developed a human mAb targeting the C-terminal domain (CTD) of PCSK9 utilizing phage display-based strategy [29]. Molecular modeling and docking techniques are widely used to predict the binding mode of a drug with its protein target [68,69].

In this work, the docking results (Figure 6E and Appendix A) showed that h5E12-L230G binds to the catalytic domain of hPCSK9. Up to fourteen amino acid residues in h5E12-L230G form the paratope that interacts with fourteen amino acid residues in hPCSK9′s epitope, creating as many as fifteen hydrogen bonds and two ionic bonds on the interaction site. These multiple interactions between the CDRs of h5E12-L230G and the catalytic domain of hPCSK9 resulted in a tight binding of h5E12-L230G to hPCSK9, highlighting the potential of h5E12-L230G as an alternative drug for hypercholesterolemia treatment.

## 5. Conclusions

In conclusion, h5E12-L230G is a humanized high-affinity anti-hPCSK9 antibody that binds to the catalytic domain of hPCSK9 with an extremely slow dissociation rate and potently inhibits PCSK9-mediated LDLR degradation, thereby greatly promoting LDL-C uptake in HepG2 cells and decreasing the serum LDL-C and TC levels in C57BL/6 mice. The data demonstrate that h5E12-L230G has the potential to serve as a therapeutic antibody targeting PCSK9 for treating hypercholesterolemia and relevant cardiovascular diseases.

## Figures and Tables

**Figure 1 biomedicines-09-01783-f001:**
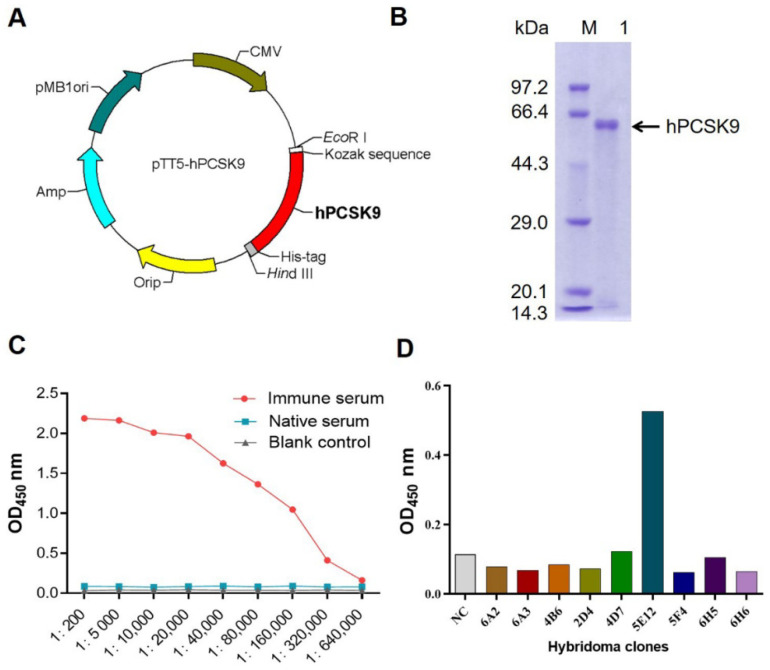
Generation of murine mAb against hPCSK9 protein. (**A**) Schematic representation of plasmid expressing hPCSK9. Kozak, Kozak consensus sequence; hPCSK9, the full-length sequence of human PCSK9 (GenBank accession number: NM_174936.3). (**B**) 10% (*w*/*v*) SDS-PAGE analysis of purified hPCSK9. M, molecular weight marker; Lane 1, purified hPCSK9 protein. There was a major band of 62 kDa corresponding to the catalytic and C-terminal domains of hPCSK9. (**C**) Serum titration after immunization with hPCSK9. The immune serum (antiserum) from mice immunized with hPCSK9 was titrated in dilutions from 1:200 to 1:640,000 and tested for antigen specificity by ELISA. Native mouse serum (pre-immune serum) was used as a negative control. (**D**) Screening of positive hybridoma clone secreting mAbs against hPCSK9 by ELISA. Cell culture medium was used as the negative control (NC).

**Figure 2 biomedicines-09-01783-f002:**
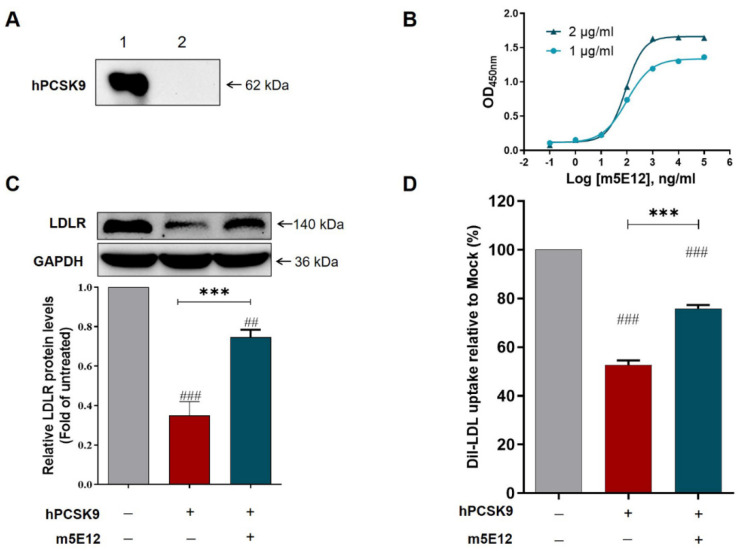
Characterization of mAb 5E12. (**A**) Western blot analysis revealed that the hPCSK9 protein (lane 1) is specifically recognized by mAb 5E12. The solvent vehicle of hPCSK9 protein (PBS buffer) was served as a negative control (lane 2). (**B**) Nonparametric test fitting curve of the results of non-competitive ELISA. The EC_50_ was equal to 81.36 ng/mL and 90.42 ng/mL when the concentrations of hPCSK9 protein were 1 μg/mL and 2 μg/mL, respectively, by which *K*_aff_ could be calculated as 1.0^4^ × 10^9^ M^−1^, using the formula of *K*_aff_ = (n − 1)/2(n[Ab’]t − [Ab]t). (**C**) Inhibitory effect of m5E12 on PCSK9-mediated LDLR degradation as assessed by Western blot analysis. HepG2 cells were treated with 20 μg/mL hPCSK9 alone or co-treated with 50 μg/mL m5E12 for 12 h, then the protein levels of LDLR were determined by Western blot. The changes of LDLR were calculated relative to that of the Mock group (vehicle control) after calibration with GAPDH in each lane. (**D**) Effect of m5E12 on PCSK9-mediated inhibition of LDL-C uptake in HepG2 cells. ^##^
*p* < 0.01 and ^###^
*p* < 0.001 vs. Mock group; *** *p* < 0.001 vs. hPCSK9 group. Data are means ± SEM of 3 independent experiments.

**Figure 3 biomedicines-09-01783-f003:**
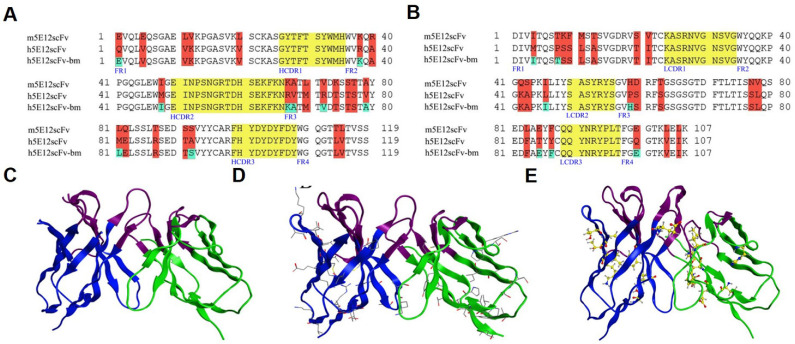
Humanization of m5E12scFv. (**A**,**B**) Sequence alignment of the VH (**A**) and VL (**B**) domain of m5E12scFv, h5E12scFv, and h5E12scFv-bm. The canonical residues back-mutated to murine residues in FRs of h5E12scFv-bm are marked in green, and the residues different between murine and humanized scFvs are marked in red. The CDRs are marked in yellow. (**C**) The homology modelled structures of m5E12scFv. The murine heavy chain (PDB ID: 10AR_H) and murine light chain (PDB ID: 1H8N_A) were used as templates for modelling of m5E12scFv. (**D**) The homology modelled structures of h5E12scFv. the homo heavy chain (PDB ID: 3HC0_H) and homo light chain (PDB ID: 1T04_A) were used as templates for modelling h5E12scFv. (**E**) The homology modelled structures of h5E12scFv-bm. The humanized heavy chain (PDB ID: 1IT9_H) and homo light chain (PDB ID: 1AD9_L) were used as templates for modelling h5E12scFv-bm. The FRs of heavy chains and light chains are presented in blue and green, respectively; the CDRs of variable regions are presented in purple; the residues in h5E12scFv different from m5E12scFv are presented in the skeletal formula; the back-mutated residues in h5E12scFv-bm are presented in the ball-and-stick formula.

**Figure 4 biomedicines-09-01783-f004:**
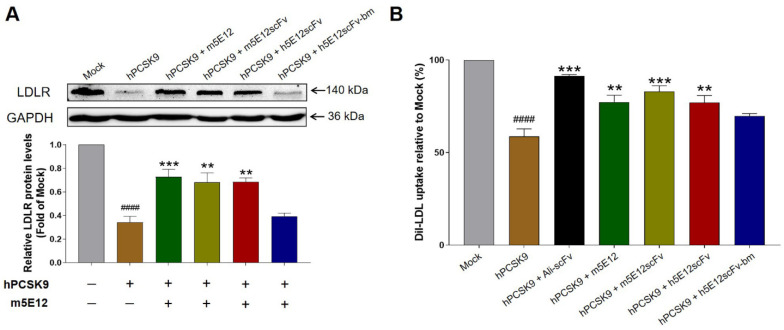
Selection of humanized 5E12 scFv. (**A**) Inhibitory effect of humanized 5E12scFv on PCSK9-mediated LDLR degradation as assessed by Western blot analysis. HepG2 cells were treated with 20 μg/mL PCSK9 alone or co-treated with 50 μg/mL m5E12 for 12 h, then the protein levels of LDLR were determined by Western blot. The changes of LDLR were calculated relative to that of the Mock group (vehicle control) after calibration with GAPDH in each lane. (**B**) Effect of humanized 5E12 scFv on PCSK9-mediated inhibition of LDL-C uptake in HepG2 cells. ^####^
*p* < 0.0001 vs. Mock group; ** *p* < 0.01, *** *p* < 0.001 vs. hPCSK9 group. Data are means ± SEM of 3 independent experiments.

**Figure 5 biomedicines-09-01783-f005:**
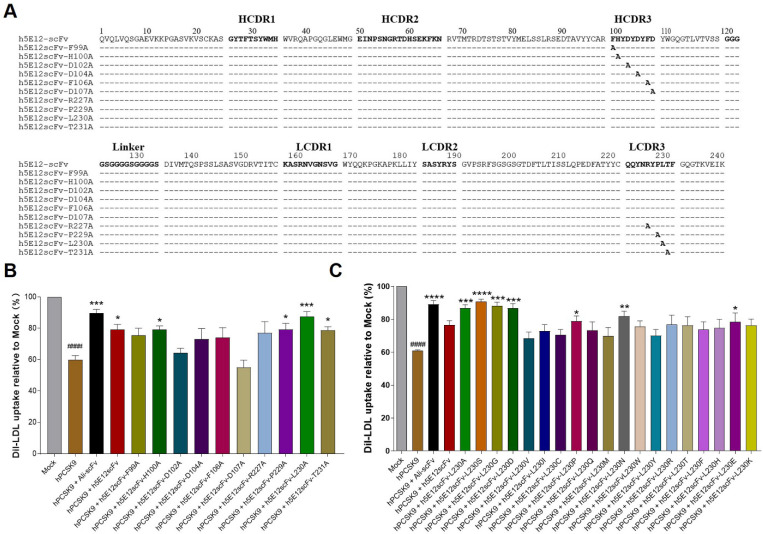
Affinity maturation and selection of h5E12scFv in vitro. (**A**) Alignment of the amino acid sequences of h5E12scFv with its corresponding alanine-scanning variants. Dashes (-) represent the same residue as h5E12scFv. (**B**) Effect of the alanine-scanning mutants of h5E12scFv on PCSK9-mediated inhibition of LDL-C uptake in HepG2. The result revealed that L230 was not necessary for maintaining h5E12scFv’s activity and its modification might improve the hPCSK9 inhibitory effect of h5E12scFv. (**C**) Effect of the saturated mutagenesis variants of h5E12scFv on PCSK9-mediated inhibition of LDL-C uptake in HepG2. The result showed that h5E12scFv-L230S was the most potent hPCSK9 inhibitor among all the h5E12scFv variants. ^####^
*p* < 0.0001 vs. Mock group; * *p* < 0.05, ** *p* < 0.01, *** *p* < 0.001, **** *p* < 0.0001 vs. hPCSK9 group. Data are means ± SEM of 3 independent experiments.

**Figure 6 biomedicines-09-01783-f006:**
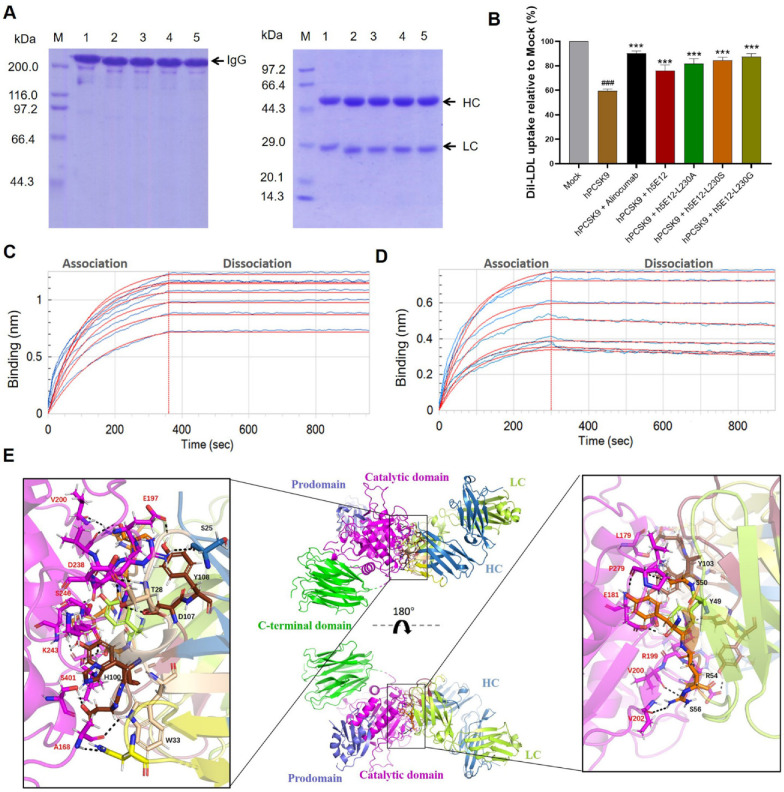
Preparation and identification of the full-length anti-PCSK9 antibodies. (**A**) SDS-PAGE analysis of purified anti-PCSK9 mAbs under non-reducing (left panel, 10% gel) and reducing (right panel, 12% gel) conditions. M, molecular weight marker; Lane 1, Alirocumab; Lane 2, h5E12; Lane 3, h5E12-L230A; Lane 4, h5E12-L230S; Lane 5, h5E12-L230G. (**B**) Effect of the anti-PCSK9 mAbs on PCSK9-mediated inhibition of LDL-C uptake in HepG2. ^###^
*p* < 0.001 vs. Mock group; *** *p* < 0.001 vs. hPCSK9 group. Data are means ± SEM of 3 independent experiments. (**C**,**D**) Kinetics measurement of h5E12-L230G (**C**) and Alirocumab (**D**) binding to surface-immobilized antigen hPCSK9 using a ForteBio Octet QK^e^ system. The biotinylated-hPCSK9 was loaded onto SA sensors and exposed to two-fold serial dilutions of antibody (800, 400, 200, 100, 50, 25, and 12.5 nM) solutions measure the association rate (k_on_), and then moved to SD buffer (PBS, pH 7.4, 0.02% Tween 20, 0.1% BSA) without antibodies measure the dissociation rate (k_off_). Blue and red lines show experimental and calculated fitting sensorgrams, respectively. Kinetic parameters were calculated by global fitting the binding curves (red lines) using Octet data Analysis software 7.1. (**E**) Intermolecular interaction analyses of the h5E12-L230G with hPCSK9. The prodomain, catalytic domain and C-terminal domain of hPCSK9 are colored in slate, magenta and green, respectively. The heavy chains’ variable regions are shown in skyblue and the light chains’ variable regions in limon. h5E12-L230G binds to the catalytic domain of PCSK9. Key residues involved in the interactions were represented as sticks models and labelled in red font for PCSK9′ key residues, black for h5E12-L230G’ key residues.

**Figure 7 biomedicines-09-01783-f007:**
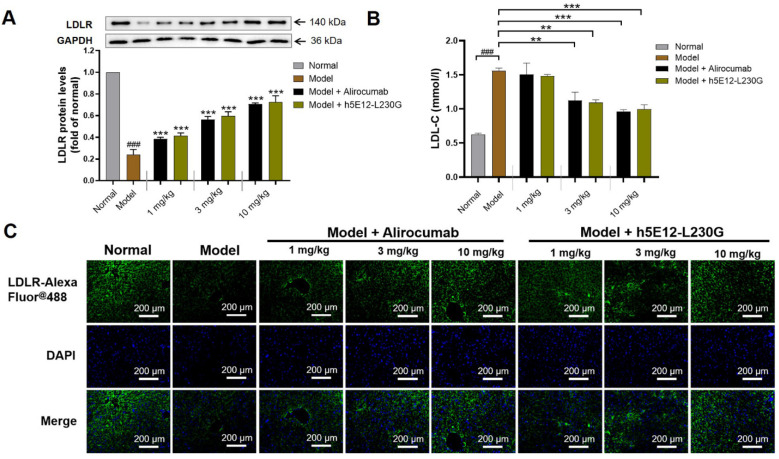
In vivo hypolipidemic efficacy of h5E12-L230G in hypercholesterolemic model mice. (**A**) Hepatic LDLR changes in h5E12-L230G or alirocumab treated C57BL/6 mice were detected by Western blot. ^###^
*p*< 0.001 vs. Normal group. ** *p* < 0.01, *** *p* < 0.001 vs. Model group. (**B**) h5E12-L230G demonstrated a significant dose-dependently LDL-C lowering effect similar to alirocumab. Results were expressed as mean ± SEM (*n* = 6 per group). See Table 1 for detailed value. (**C**) Hepatic LDLR changes in h5E12-L230G or alirocumab treated C57BL/6 mice were detected by Immunofluorescence. Scale bars = 200 μm. Data are representative of 3 independent experiments with similar results.

**Table 1 biomedicines-09-01783-t001:** Effects of h5E12-L230G on serum lipids levels in hyperlipidemic mice model.

Group		Dose (mg/kg)	LDL-C (mmol/L)	TC (mmol/L)	HDLC (mmol/L)	TG (mmol/L)
Normal group		/	0.624 ± 0.021	3.642 ± 0.161	1417 ± 0.042	0.0921 ± 0.028
Model group ^1^		/	1560 ± 0.037 ^###^	5.250 ± 0.212 ^###^	1340 ± 0.036	1351 ± 0.049 ^###^
Alirocumab group	Low dose	1	1506 ± 0.164	4.903 ± 0.071	1275 ± 0.066	1320 ± 0.052
	Medium dose	3	1122 ± 0.122 **	4650 ± 0.066 **	1417 ± 0.088	1211 ± 0.048
	High dose	10	0.964 ± 0.024 ***	4276 ± 0.062 ***	1287 ± 0.054	1162 ± 0.037 *
h5E12-L230G group	Low dose	1	1480 ± 0.0262	4.770 ± 0.045 *	1462 ± 0.084	1312 ± 0.026
	Medium dose	3	1093 ± 0.0395 **	4658 ± 0.022 **	1333 ± 0.071	1196 ± 0.048
	High dose	10	0.996 ± 0.0644 ***	4469 ± 0.063 ***	1257 ± 0.044	1155 ± 0.037 *

^1^ The model mice exhibited a serum hPCSK9 level of (1152 ± 75) ng/mL determined by an ELISA Kit for human PCSK9 (Cat^#^ E0307h, Eiaab Science Inc., Wuhan, China). * *p* < 0.05, vs. Model group. ** *p* < 0.01, vs. Model group. *** *p* < 0.001, vs. Model group. ^###^
*p* < 0.001, vs. Normal group.

## Data Availability

All data generated or analyzed during this study are included in this manuscript and Appendix A.

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
