# Peer review of "Generation of a Novel High-Affinity Antibody Binding to PCSK9 Catalytic Domain with Slow Dissociation Rate by CDR-Grafting, Alanine Scanning and Saturated Site-Directed Mutagenesis for Favorably Treating Hypercholesterolemia"

_biomedicines, 2021, doi:10.3390/biomedicines9121783_

Round 1

Reviewer 1 Report

In this study, authors generated a novel PCSK9 antibody with high binding affinity and slow dissociation rate and validated its function in regulating LDLR and LDL-c in HepG2 cells as well as in a mouse model with exogenous expression of hPSCK9. Although antibodies against PCSK9 have been approved to treated patients with primary hypercholesterolemia and stain intolerance, to generate a more effective antibody against PCSK9 remains important. Similar to authors’ recent publication (EBioMedicine . 2021 Mar;65:103250), in the present manuscript, authors reported an another antibody against PCSK9 with high efficiency in increasing LDLR expression and LDL uptake. However, crude manuscript preparation limits the importance of this study.  

  1. Authors should explain the relationship between Table 1 of this manuscript and table 1 published in EBioMedicine 2021. Some values, such as TG levels, are identical in these two study.

.

  1. What is FAP2M21 in table 1?

  1. The serum level of PSCK9 in the mouse model with intravenous injection of expression vector pTT5-hPSCK9 should be included in table 1.

  1. Authors’ recent publication with developing a humanized anti-PCSK9 antibody through phage display-based strategy should be included in the discussion section (EBioMedicine. 2021 Mar;65:103250).

5. It is suggested to reduce the number of references in the manuscript. 

Author Response

Response to Reviewer 1 Comments

Point 1:  Authors should explain the relationship between Table 1 of this manuscript and table 4 published in EBioMedicine 2021. Some values, such as TG levels, are identical in these two studies.

Response:

We are terribly sorry for the crude preparation of this manuscript, especially for Table 1.

Just as you thought, there are some relationships between Table 1 of this manuscript and table 4 published in EBioMedicine 2021 for the reason that the data in the two tables were derived from the same experiment. In previous trials, we simultaneously evaluated the hypolipidemic efficacy of h5E12-L230G (in this manuscript) and FAP2M21 (in our recent publication EBioMedicine. 2021 Mar; 65:103250) using the same control groups (Normal group, Model group and positive control (Alirocumab) group). Thus, only the lipid values in the control group are identical in these two studies.

However, we must admit and clarify that the values in Table 1 of this manuscript are disordered and wrongly presented due to our carelessness. As you can see in the unrevised manuscript, the values in Table 1 are inconsistent with Figure 7B and the result description in section 3.7 as well as relevant patent (CN. Pat. No. 110981962A), while the data in Figure 7B, result 3.7 and the patent are consistent with each other. What's more, the LDL-C levels are even higher than total cholesterol (TC) levels in Table 1. All these extremely abnormal and incompatible data in Table 1 just indicate that these errors are unintentionally caused by our careless preparation of Table 1. We deeply apologize for our carelessness and sincerely appreciate your meticulous review of our manuscript.

We have accordingly carefully amended Table 1 in the revised manuscript (See Table 1, page 15). We want to declare here that the values in revised Table 1 are correctly derived from the raw data and consistent with the corresponding figure and description in the original manuscript as well as the data in the related patent (CN. Pat. No. 110981962A).

Point 2: What is FAP2M21 in table 1?

Response:

FAP2M21 in Table 1 should be revised to h5E12-L230G.

As described above, this mistake was caused by our crude preparation of Table 1, and we’ve corrected it in the revised manuscript (See Table 1, page 15).

Point 3: The serum level of PCSK9 in the mouse model with intravenous injection of expression vector pTT5-hPSCK9 should be included in table 1.

Response:

Unlike gene-silencing agents work by inhibiting PCSK9 synthesis and reducing serum PCSK9 levels, PCSK9 antibodies regulate LDL-C levels by blocking the interaction of PCSK9 and LDLR (Marais D A. et al. Curr Opin Lipidol, 2012, 23(6):511-517; Ni-Ya H E. et al. Acta Pharmacologica Sinica, 2017, 38(003):301-311). Thus, consistent with several similar articles (Chan. et al. PNAS, 2009, 106:9820-9825; Weider E, et al. J Biol Chem, 2016, 291(32).), we did not systematic evaluate the serum PCSK9 levels in each group after treatment.

To verify whether human PCSK9 (hPCSK9) was overexpressed in the mouse model, we randomly selected three model mice on one day before the treatment (on day 6) and tested the serum hPCSK9 levels by an ELISA Kit for Human Proprotein convertase subtilisin/kexin type 9 (Cat# E0307h, Eiaab Science Inc, Wuhan). The results showed that serum hPCSK9 level in the model mice was as high as (1152 ± 75) ng/ml.

Based on your suggestion, we added PCSK9 levels in Table 1 in the revised manuscript (See Table 1, page 16) as follows:

1The model mice exhibited a serum hPCSK9 level of (1152 ± 75) ng/ml determined by an ELISA Kit for human PCSK9 (Cat# E0307h, Eiaab Science Inc, Wuhan, China).”

Point 4: Authors’ recent publication with developing a humanized anti-PCSK9 antibody through phage display-based strategy should be included in the discussion section (EBioMedicine. 2021 Mar;65: 103250).

Response:

We already included this article in the discussion section in the revised manuscript (See page 17, line 852-854) as follows:

“We have also previously developed a human mAb targeting the C-terminal domain (CTD) of PCSK9 utilizing phage display-based strategy [29]”

Point 5: It is suggested to reduce the number of references in the manuscript.

Response:

As you suggested, we already reduced the number of references in the revised manuscript.

Reviewer 2 Report

I rate the manuscript submitted to the "Biomedicines" journal by Zhengli Bai and co-authors very highly.
The study results indicate that h5E12-L230G is a highly potent antibody binding to PCSK9 catalytic domain with slow dissociation rate which may be utilized as a therapeutic candidate for treating hypercholesterolemia and relevant cardiovascular diseases.
The results of the study are promising and may indicate an interesting therapeutic option for civilization diseases.
The study was carried out correctly.
My only remarks concern the construction of the manuscript itself.
A separate sub-chapter on the objectives of the study should be distinguished. You need to specify the main goal, but also the specific goals of the research.
The research methodology was described in too much detail. Some of the presented data can be omitted by referring to the appropriate references.
I would transfer less important data to the supplement.
More space should be devoted to the discussion. The discussion of the results in the current version of the manuscript is too general. The possible use of the results in future research and in future clinical practice should be discussed more broadly.

Author Response

Response to Reviewer 2 Comments

Point 1:  My only remarks concern the construction of the manuscript itself.
A separate sub-chapter on the objectives of the study should be distinguished. You need to specify the main goal, but also the specific goals of the research.

Response:

Thank you for your instructive suggestions, we have rewritten the introduction section to specify the goal of this study and also listed a separate sub-chapter on the objective of the study as follows:

“In the present study, we aimed to generate a high-affinity hybridoma-derived mAb against hPCSK9 for future therapeutic applications. In order to reach this goal, a CDR-grafting humanization approach was initially performed to reduce the immunogenicity of selected murine mAb, some key residues in murine FRs that might influence the antigen-binding activity were back-mutated attempting to restore full affinity. Thereafter, alanine-scanning mutational analysis followed by a saturated site-directed mutagenesis process were conducted on the critical amino acid residue of humanized antibody to further improve its affinity. Finally, the selected optimized scFvs were reformatted into full-length IgG by fusing with modified human IgG1 constant region for favorably treating hypercholesterolemia in vivo.”

Point 2:  The research methodology was described in too much detail. Some of the presented data can be omitted by referring to the appropriate references.

Response:

We already simplified the research methodology in the revised manuscript (See section 2.3, 2.4 on page 3, and section 2.6 on page 4).

Point 3:  I would transfer less important data to the supplement.

Response:

We already transferred some less important data (Figure 1C, Figure 1F, Figure 4A, and Figure 4B) to the supplement in the revised manuscript.

Point 4:  More space should be devoted to the discussion. The discussion of the results in the current version of the manuscript is too general. The possible use of the results in future research and in future clinical practice should be discussed more broadly.

Response:

We already rewrote and deepen the discussion in the revised manuscript (See section 4 on pages 16, 17).

Round 2

Reviewer 1 Report

I have no comment. This manuscript is acceptable.